# Comparable Vδ2 Cell Functional Characteristics in Virally Suppressed People Living with HIV and Uninfected Individuals

**DOI:** 10.3390/cells9122568

**Published:** 2020-12-01

**Authors:** Matthew L. Clohosey, Brendan T. Mann, Paul L. Ryan, Tatiyana V. Apanasovich, Sanjay B. Maggirwar, Daniel J. Pennington, Natalia Soriano-Sarabia

**Affiliations:** 1UNC-HIV Cure Center, Department of Medicine, University of North Carolina at Chapel Hill, Chapel Hill, NC 27009, USA; matthew_clohosey@med.unc.edu; 2Department of Microbiology, Immunology, and Tropical Medicine, George Washington University, Washington, DC 98092, USA; bmann@gwmail.gwu.edu (B.T.M.); smaggirwar@email.gwu.edu (S.B.M.); 3Centre for Oral Immunobiology and Regenerative Medicine, Institute of Dentistry, Barts and The London School of Medicine and Dentistry, Queen Mary University of London, London E1 2AT, UK; p.l.ryan@qmul.ac.uk; 4Department of Statistics, George Washington University, Washington, DC 98092, USA; apanasovich@gwu.edu; 5Centre for Immunobiology, Blizard Institute, Barts and The London School of Medicine and Dentistry, Queen Mary University of London, London E1 2AT, UK; d.pennington@qmul.ac.uk

**Keywords:** gammadelta T cells, HIV, antiretroviral therapy, cytotoxicity, TCR

## Abstract

Crosstalk between innate and adaptive pathways is a critical component to developing an effective, lasting immune response. Among natural effector cells, innate-like γδ T cells promote immunity by facilitating communication between the two compartments and exerting cytotoxic effector functions. Dysregulation of γδ T cell populations is a byproduct of primary human immunodeficiency virus (HIV) infection. This is most pronounced in the depletion and loss of function within cells expressing a Vγ9Vδ2 TCR (Vδ2 cells). Whether or not prolonged viral suppression mediated by antiretroviral therapy (ART) can reverse these effects has yet to be determined. In this study, we present evidence of similar Vδ2 cell functional responses within a cohort of people living with HIV (PLWH) that has been stably suppressed for >1 year and uninfected donors. Through the use of aminobisphosphonate drugs, we were able to generate a comprehensive comparison between ex vivo and expanded Vδ2 cells within each group. Both groups had largely similar compositions of memory and effector phenotypes, post-expansion TCR repertoire diversity, and cytotoxic capabilities. Our findings support the notion that ART promotes the recovery of Vδ2 polyfunctionality and provides insight for strategies aiming to reconstitute the full immune response after infection with HIV.

## 1. Introduction

In contrast to conventional αβ T lymphocytes, γδ T cells are distinguished by the expression of a T cell receptor (TCR) comprised of heterodimeric γ and δ chains as well as recognition of non-peptide antigens outside of classical major histocompatibility complex (MHC) presentation [1,2]. A subset expressing the Vγ9Vδ2 TCR, herein referred to as Vδ2 cells, comprise up to 15% of circulating CD3+ lymphocytes within the peripheral blood of healthy individuals. Vδ2 cells recognize low molecular weight phosphorylated intermediates of isoprenoid biosynthesis, or phosphoantigens (P-Ag), such as isopentyl pyrophosphate (IPP) and (E)-4-hydroxy-3-methyl-but-2-enyl pyrophosphate (HMBPP) [3,4]. Treatment with aminobisphosphonate drugs (N-BPs), such as pamidronate (PAM) or zoledronate (ZOL), results in the accumulation of IPP and subsequent induction of Vδ2 cell activation [5,6]. Activated Vδ2 cells exert potent cytotoxic capabilities and promote a strong T_H_1 response to a number of diverse pathogens and tumor types [7,8,9,10,11]. Cytotoxic effector subsets are distinguished by early expression of the natural killer cell marker CD56 [12]. Additionally FcγRIII (CD16) expressing Vδ2 cells possess anti-HIV properties [13], and may play a role in controlling the frequency of latently infected cells [14]. These attributes make Vδ2 cells an attractive therapeutic candidate for innovative strategies targeting persistent HIV infection [14,15] as well as both solid and hematological malignancies [16]. Clinical trials of γδ T cell therapies for cancer treatment were generally well tolerated but demonstrated variable efficacy, most likely due to interpersonal differences in Vδ2 cell population frequencies. The clinical significance of these differences remains unclear [17,18].

Characterizing the phenotypic differences of Vδ2 cells may provide insights beneficial to adoptive transfer strategies or targeted therapies. We (P.L.R. and D.J.P) have previously identified four phenotypically distinct Vδ2 cell populations that vary in frequency between healthy donors [19]. These populations are defined as CD28+CD27+CD16- (γδ^(28+)^), CD28-CD27+CD16- (γδ^(28−)^), CD28-CD27-CD16- (γδ^(16−)^), and CD28-CD27-CD16+ (γδ^(16+)^) with γδ^(28+)^ and γδ^(16+)^ being the immunodominant Vδ2 cell proliferative and cytotoxic responses respectively. Further investigation of these populations in the context of individual diseases is still needed. Specifically, primary HIV infection directly causes dysregulation of the natural frequencies, polyfunctionality, and TCR repertoire of γδ T cells [20]. Whether or not prolonged antiretroviral therapy (ART) is capable of restoring these properties has yet to be resolved. The number of circulating Vδ2 cells as well as cytotoxic function and cytokine production remain dampened within the first six months of ART regardless of early intervention [21]. However, a previous study conducted by our group showed that after one year of suppressive ART, Vδ2 cells were capable of killing latently infected CD4 T cells despite continued depression of overall frequencies [22]. Additionally, our understanding of the effects of ART on the Vδ2 TCR repertoire remains limited. Individuals on prolonged ART show only partial recovery highlighted by an increase in TCR diversity but sustained unresponsiveness to P-Ag [23]. We hypothesized that despite continued depletion of Vδ2 cells in ART-treated individuals, the phenotypic profiles, TCR repertoire, and cytotoxic capabilities are comparable to those observed in uninfected individuals. We conducted comprehensive comparisons of Vδ2 cells phenotype and function between donors from these two groups. Our study shows that, overall, these Vδ2 cell characteristics in PLWH on suppressive ART for >1 year are largely similar to uninfected donors.

## 2. Materials and Methods

### 2.1. Cells and Cell Lines

Characteristics of the HIV participants included in this study have been previously reported [14,22]. All PLWH were on stable ART with plasma HIV-1 RNA < 50 copies/mL for >1 year before enrollment. All participants provided written informed consent and studies were approved by the University of North Carolina’s Institutional Review Board (IRB). Buffy coats from uninfected donors were obtained from the New York Blood Center (Long Island City, NY, USA) without any identifiers. Daudi human Burkitt’s lymphoma cell line was obtained from the American type culture collection (ATCC).

### 2.2. Vδ2 Cell Expansion

PBMCs from ART-suppressed PLWH or uninfected donors were thawed, counted, and rested overnight. An aliquot was expanded, and the rest was used for phenotypic analysis by flow cytometry. Five million PBMCs were cultured in media supplemented with 10% FBS and 2.5 μg/mL of PAM (Sigma-Aldrich, St. Louis, MO, USA) and 100 U/mL of IL-2 (Peprotech, Rocky Hill, CT, USA) for 7 days. Media containing 100 U/mL IL-2 was refreshed at day 3. After seven days of expansion, cells were harvested, counted, and resuspended in staining buffer (i.e., 2% FBS in PBS) to perform flow cytometry assays. Both baseline and expanded cells were stained using the same three monoclonal antibody panels.

### 2.3. Vδ2 Cell Phenotypic Analysis

PBMCs were stained using combinations of monoclonal antibodies in three different panels. For the baseline phenotypic analysis, approximately 40 million PBMCs were enriched for γδ T cells using a commercially available kit (StemCell Technologies). After expansion, cells were washed and stained using the same panels for the baseline analysis. The following monoclonal antibodies (mAbs) were used: CD27 (clone M-T271), CD3 (clone SK7), CD28 (clone CD28.2), CD16 (clone 3G8), CD69 (clone FN50), HLA-DR (clone L243), Vδ2 (clone M-T271), CD56 (clone HCD56), and PD-1 (clone EH12.2H7) (all from Biolegend, San Diego, CA, USA except CD3 from BD Biosciences, San Jose, CA, USA). Cells were incubated in staining buffer with appropriate mAbs for 20 min on ice in the dark, washed, and fixed in 2% paraformaldehyde (PFA) solution. In some experiments, total PBMCs and γδ T cell enriched PBMCs were compared in parallel to determine the potential for bias from the enrichment procedure in our baseline populations. This included an intracellular staining for the proliferative marker Ki-67 (clone Ki-67). Cells were fixed and permeabilized using Cytofix/Cytoperm solution (BD Biosciences, San Diego) after extracellular staining. The permeabilized cells were then washed, incubated in staining buffer with the Ki-67 mAb on ice in the dark, then washed again to remove any unbound protein. Stained cells were acquired on the Attune NxT (Life Technologies, Carlsbad, CA, USA) and analyzed using FlowJo v10.1.

### 2.4. Cytokine Production Analysis

Cytokine production was measured using a previously reported method [24]. Supernatant from PBMC cultures incubated with 2.5 μg/mL PAM and 100 U/mL IL-2 for seven days were analyzed by ELISA (Advanced Bioscience Laboratories, Rockville, MD). After the expansion, supernatants from the 7-day expansion were harvested and stored at −80 °C until further cytokine ELISA measurement was performed. IFN-γ, TNF-α, and Granzyme (Grz) B were measured both in cultures from PLWH and uninfected donors, following the manufacturer’s instructions.

### 2.5. Degranulation and Cytotoxicity

PBMCs derived from PLWH or uninfected donors, were incubated with PAM and IL-2 for 7 days and Vδ2 cells were FACS-sorted using a previously reported strategy [25]. Purity of the isolated populations was >99.9%. Sorted cells were rested overnight in the presence of 30 U/mL IL-2 and then CD107a (Lamp1) antibody was added to the culture in the presence of Brefeldin A (Biolegend). Daudi Burkitt’s lymphoma cells lack MHC expression and are highly susceptible to Vδ2 cell-mediated killing [26]. Daudi cells were cultured alone or added to the culture at a 1:1 (target:effector) ratio for 4 h. Cells were washed and stained for surface markers CD3 and Vδ2 for 20 min on ice. After fixation and permeabilization (BD biosciences) cells were stained for intracellular markers GrzB and perforin for 30 min. To analyze Daudi cell killing, the cells were incubated for 15 min with viability dye 7-AAD (BD Biosciences) prior to the flow cytometry analysis on the Attune NxT (Life Technologies, Carlsbad, CA, USA) and analyzed using FlowJo v10.1.

### 2.6. Immune Repertoire Sequencing Assays

One million basal and expanded PBMC from PLWH and uninfected individuals were pelleted and stored at −80 °C. Total RNA was extracted using the Qiagen total RNA isolation mini kit (Qiagen) and concentration and purity analyzed using the Nanodrop spectrophotometer (Thermofisher). The genomic libraries were made following Archer VariantPlex Protocol for Illumina (Archer, Cat: SK0096) which utilizes gene-specific primers for a targeted amplification to identify mutations and is suited for low DNA input. This protocol was used in conjunction with the corresponding target enrichment panel utilizing molecular barcode adapters (MBCs). The kit employs fragmentation by thermocycler, followed by adding target panel MBCs which tag each DNA molecule with a barcode and common region, followed by adapter ligation and two steps of PCR amplification. Samples were then multiplexed together to increase sample diversity before running on a NextSeq 500 (requiring a spike in of 5% PhiX to increase the library diversity). The samples were identified through both the molecular tag (added during adapter ligation through the MBC Adapters with the p5/i5 index) and the barcode/Illumina Index 1 (added during the second PCR cycle with the p7/i7 index) and were analyzed using Archer Analysis software.

### 2.7. Statistical Analysis

Data obtained were analyzed using nonparametric methods. Different groups were compared by Mann–Whitney U test and repeated measures within the same groups were analyzed by Wilcoxon signed-rank test (paired samples). Fisher’s exact test was used to compare racial distribution between profiles and Kruskal–Wallis test to compare cell distribution among different profiles. Finally, Spearman’s rank-order correlation was used to analyze associations between two different continuous variables. In addition, we used a ranked based version of a mixed effect ANOVA to account for random subject effects. The thresholds for *p*-values were corrected for multiple testing. Graphs were created using GraphPad Prism v.8.4.2 (GraphPad Software, San Diego, CA, USA) and statistical analyses were performed in the R language and environment for statistical computing (R Development Core Team).

## 3. Results

### 3.1. Baseline Characteristics

The study included 23 uninfected donors and 20 PLWH that had been on stable ART, with plasma viral load <50 copies/mL for at least one year (Table 1). Vδ2 cells are profoundly depleted in HIV infection [22,27]. In the present study, we found a mean of 0.4% Vδ2 cells within total CD3+ lymphocytes in ART-suppressed PLWH compared to 6.8% in uninfected donors (Appendix A). In order to more accurately analyze subpopulations within Vδ2 cells of PLWH, we performed a γδ T cell enrichment prior to baseline phenotypic characterization. The enrichment step did not affect the phenotype as evidenced by similar expression frequencies of CD27 and CD28 (Appendix A), activation markers CD69, HLA-DR and proliferation marker Ki67 (Appendix A) between ex vivo isolated PBMCs and enriched PBMCs. These results indicate a lack of selective enrichment or depletion of certain populations.

### 3.2. Comparable Expression of CD16 and CD56 on Vδ2 Cells from Uninfected Donors and PLWH on Stable ART

We investigated differences in cytotoxic markers CD16 and CD56 expression in Vδ2 cells between uninfected donors and PLWH, and whether expansion with PAM and IL-2 altered the distribution. CD16 and CD56 expression on ex vivo Vδ2 cells were similar between uninfected donors and PLWH (Figure 1A, *p* = 0.41 for CD16 and *p* = 0.59 for CD56, Mann–Whitney U-tests). Upon expansion with PAΜ and IL-2, CD16 and CD56 expression were upregulated in both uninfected and PLWH (Figure 1A,B, respectively, *p* < 0.001, Wilcoxon paired signed-rank tests) showing that Vδ2 cells’ capacity to upregulate either cytotoxic marker was not impaired in stable durably suppressed PLWH.

We next analyzed whether expression of these markers was different according to race in PLWH (African American (AA) compared to Caucasian (CC), since only one individual was Hispanic). AA donors had a mean of 0.3% Vδ2 cells within total CD3+ lymphocytes compared to CC donors that had 0.5% (*p* = 0.89, Mann–Whitney U-test). Race information was not available for the uninfected group. Data on CD16 expression was available in nine AA and ten CC donors, and CD56 expression was available in six AA and ten CC donors. Both CD16 and CD56 expression was similarly distributed in ex vivo and expanded Vδ2 cells. Mean baseline CD16 expression in AA was 8.2% compared to 11.7% in CC, while mean CD16 expression on expanded Vδ2 cells was 27.1% in AA compared to 33.4% in CC (Figure 1C, *p* = 0.42 and *p* = 0.43, respectively, Mann–Whitney U-test). Mean baseline CD56 expression was 28.4% in AA compared to 17.8% in CC, while mean expression on expanded Vδ2 cells was 45.2% in AA, compared to 43.7% in CC (Figure 1C, *p* = 0.59 and *p* = 0.63, respectively, Mann Whitney U-test).

### 3.3. Similar Composition of Vδ2 Cell Populations in Uninfected Donors and PLWH

Memory and effector Vδ2 cell populations among PBMCs were defined as previously described [19] γδ^(28+)^ were defined as [CD28+CD27+CD16-], γδ^(28−)^ were [CD28-CD27+CD16-], γδ^(16−)^ expressed [CD28-CD27-CD16-], and finally γδ^(16+)^ were [CD28-CD27-CD16+]. Baseline Vδ2 cell populations were comparable between uninfected donors and PLWH (Figure 2A). To analyze whether expansion alters Vδ2 cell phenotype composition, PBMCs from PLWH and uninfected donors were treated with PAM and IL-2 for seven days. In PLWH the frequency of the γδ^(28+)^ population decreased while the frequency of the other three populations γδ^(28−)^, γδ^(16−)^, and γδ^(16+)^ increased (Figure 2B), and the pattern was similar in PLWH treated in acute or chronic phases of the infection (Figure 2C). Uninfected individuals showed a similar pattern to PLWH except for the γδ^(28−)^ population that maintained a constant frequency after seven days of expansion (Figure 2D). Incubation with PAM and IL-2 for 7 days induced a comparable fold-change expansion between PLWH and uninfected donors, showing the greatest impact on the CD27-CD28- effector memory populations, and especially in the cytotoxic CD16+ subpopulation (Figure 2E).

### 3.4. Vδ2 Cell Populations Are Distributed amongst Several Profiles in Both Uninfected Donors and PLWH

According to the relative distribution of effector/memory Vδ2 cell populations, healthy individuals can be classified into six different functional groups or profiles [19]. Each profile is distinguished by the relative proportions of Vδ2 cell subpopulations (γδ^(28+)^, γδ^(28−)^, γδ^(16−)^, γδ^(16+)^). Profiles 1–4 are dominated by γδ^(28+)^ and γδ^(28−)^ populations. This is contrasted by profile 5 which has a fairly equal representation of γδ^(16−)^ and γδ^(16+)^ and profile 6 which has a single dominant population defined as γδ^(16+)^. Differences in subpopulation frequencies were also shown to be predictive of functional responses to antigen stimulation. Donors with a pre-existing γδ^(28+)^ immunodominant profile demonstrated substantial proliferation compared to donors possessing a profile marked by a higher γδ^(16+)^ population. Instead, the latter were shown to have significantly higher cytotoxic potential. The distribution of our cohort of 23 uninfected individuals was similar to what has been previously reported, although less prevalent profiles five and six were lacking. This difference is most likely due to the smaller sample size of the current study (Figure 3A). Similarly, Vδ2 cell populations from PLWH were classified into four distinct profiles (Figure 3B). In PLWH, age, sex, initiation of ART in acute or chronic HIV infection and time on ART (*p* > 0.05, not shown), did not influence the distribution of Vδ2 cell populations. Interestingly, although overall expression of CD16 on Vδ2 cells was similar between AA and CC PLWH donors (Figure 1C), AA donors were most likely grouped in profile 1 while CC were mostly grouped in profile 2 (Figure 3C, *p* = 0.02, Fisher exact test), with one Hispanic individual classified in Profile 3. In addition, CD4 T cell count at the time of the study or CD4 T cell count nadir were not associated with Vδ2 cell populations’ distribution (Figure 3D, *p* = 0.95 and *p* = 0.28, respectively, Kruskal–Wallis rank sum test). CD8 T cell count had a marginal effect and individuals with higher CD8 T cell frequency of were preferentially grouped in profile 2 (Figure 3D, *p* = 0.07, Kruskal–Wallis rank sum test).

### 3.5. Cytokine Release after Vδ2 Cell Expansion

We analyzed total levels of T_H_1 cytokines IFN-γ and TNF-α, and GrzB released in the supernatants of cell cultures after seven days of expansion with PAM and IL-2. In PLWH, levels of IFN-γ were higher while levels of TNF-α and GrzB were lower compared to uninfected donors (Figure 4A–C). In addition, in PLWH but not in uninfected donors, GrzB production positively correlated with IFN-γ release (Figure 4D, *p* = 0.037). the expression of CD56 on expanded Vδ2 cells positively correlated with secreted GrzB levels (Figure 4E, *p* = 0.005). In addition, we analyzed the frequency of NK cells present in the culture that could be contributing to GrzB production. Although they were present at variable frequencies, NK cell proportion did not correlate with GrZB production (Appendix A, *p* = 0.06), suggesting that the majority of this protease was produced by Vδ2 cells. A correlation was not found between secreted cytokines and GrzB and the total number of Vδ2 cells or CD16 expression (not shown, *p* > 0.05 in all cases).

### 3.6. Cytolytic Capacity and Vδ2 Cell-Mediated Killing

Vδ2 cells were isolated after ex vivo expansion with PAM and IL-2 and cocultured with Daudi cells at a 1:1 ratio. Degranulation capacity and intracellular production of perforin and GrzB was measured in six PLWH and four uninfected individuals (Appendix A). Similar expression of CD107a and production of perforin was found between both groups (Figure 5A,B). These markers increased similarly in both PLWH and uninfected individuals upon coculture with Daudi cells (Figure 5A–C). GrzB production was initially higher in isolated Vδ2 cells from PLWH than in uninfected donors after expansion (Figure 5C), but production was comparable upon coculture with Daudi cells (Figure 5C).

In addition, since Vδ2 cells from AA PLWH were mostly categorized as central memory cells, we analyzed whether expanded Vδ2 cells’ cytotoxic function in AA donors was diminished compared to their CC counterparts. In limited analyses using samples from five individuals, race-dependent differences in degranulation capacity or intracellular production of perforin or GrzB were not detected (Figure 5D). CD107a expression positively correlated with both perforin and GrzB production (Figure 5E,F). Finally, expanded Vδ2 cells from PLWH were able to effectively lyse Daudi cells (Appendix A) in comparable fashion to uninfected donors (Figure 5G). We did not detect differences in Daudi cell killing according to race in PLWH, although we could only include four individuals in these analyses (Figure 5H).

### 3.7. TCR Repertoire Analysis

Sequencing of TCRγ transcripts from PLWH patients on stable and suppressive ART for >1 year did not reveal any obvious selection for particular Vγ-regions when compared with uninfected controls (Figure 6A). Moreover, activation of PBMCs with PAM and IL-2 led to preferential expansion of Vγ9 (that pairs with Vδ2 in phosphoantigen-responsive γδ T cells) sequences in both groups (Figure 6A). Evaluation of shared Vγ9 sequences within each group and between each group revealed that each individual shared Vγ9 sequences with at least four other individuals from the total cohort (Figure 6B). Thus, PLWH retain Vδ2 cells that use “public” Vγ9 sequences that have previously been identified as being present in multiple individuals in the general population.

Analysis of Vγ9 sequences in ex vivo PBMCs revealed that PLWH may possess reduced repertoire diversity compared to uninfected donors as measured by TCR occupancy (Figure 6C). Although our limited sample size of uninfected donors limits making definitive conclusions from this stage of the analysis. In contrast, PBMCs treated with PAM and IL-2 showed broadly similar patterns in all eight individuals, with no evidence of skewness towards either restricted or diverse repertoires in PLWH, suggesting expansion of a similar range of Vγ9 clones (Figure 6D).

Finally, analysis of Vγ9 CDR3 nucleotide length showed that samples from ex vivo PLWH samples had shorter CDR3s when compared with samples from uninfected donors, although no particular CDR3 length size was selected (Figure 6E). Interestingly, after PAM and IL-2 exposure, the pattern of CDR3 length was similar for samples from uninfected donors and PLWH, with preferential expansion of CDR3s with nucleotide length of 48 and to a lesser extent lengths 46, 52, and, 54 (Figure 6E).

## 4. Discussion

Vδ2 cells are key contributors to immune crosstalk that drives the T_H_1 cell-mediated responses needed for clearance of microbial infections [28]. HIV infection specifically causes the depletion and dysregulation of circulating Vδ2 cells [25,27,29]. It is clear that treatment is unable to reconstitute the overall number of Vδ2 cells, but determining the capacity of ART to restore their functionality is a contentious subject [30]. In this study, we have compared the phenotype, cytotoxic capacity, and TCR repertoire of Vδ2 cells from uninfected donors and PLWH on suppressive ART for >1 year. Although numerically reduced, Vδ2 cells from PLWH and uninfected donors had comparable: (1) ex vivo frequencies of proliferative and cytotoxic Vδ2 cell subpopulations, (2) clonal responses to PAM + IL-2 (3) degranulation and cytotoxic function. Our results indicate that despite sustained depletion in treated infection, PLWH on continuous and suppressive ART possess comparable Vδ2 cell phenotypes and functionality than uninfected individuals.

HIV-mediated depletion of Vδ2 cells preferentially affects those expressing a Vγ9-Jγ1.2 TCR leaving behind a repertoire that shows attenuated responsiveness to P-Ag stimulation [23,31,32]. This is evident in the diminished in vitro expansion following P-Ag stimulation, as previously reported by our group and others [22,33]. In contrast, Vδ2 cells in PLWH retain the ability to expand in the presence of N-BPs, such as PAM and ZOL, similar to uninfected donors [34,35]. Our results show that PLWH may possess less diverse Vδ2 cell TCR repertoires compared to uninfected donors ex vivo, but similar induced clonality as a result of expansion with PAM + IL-2 was observed. The presence of antigen driven activation raises the possibility that ART may recover some Vδ2 cell functionality in fully suppressed individuals, although a longitudinal study in PLWH is required to confirm this finding.

We (P.L.R and D.J.P) previously characterized Vδ2 T cell composition in healthy individuals showing a heterogeneous distribution of CD27, CD28 and CD16 expression, and defined distinct proliferative and cytotoxic phenotypic profiles based on these distributions [19]. In this study, we analyzed Vδ2 cell phenotype and found that levels of CD16 and CD56 on Vδ2 cells were comparable between uninfected donors and PLWH. CD16+ Vδ2 cell populations are associated with poor proliferative responses, control of HIV infection, and elimination of latently HIV-infected cells [13,14,22]. Using previously defined Vδ2 cell phenotype classification [19], our results show a similar composition and distribution of Vδ2 cell population profiles in uninfected donors and PLWH. After one week of ex vivo stimulation with PAM and IL-2, the frequency of γδ^(28+)^ cells from PLWH decreased while the γδ^(28−)^, γδ^(16−)^ and γδ^(16+)^ populations increased. These results demonstrate that Vδ2 cell profile distribution and ability to upregulate cytotoxic markers is not impaired in PLWH on extended suppressive ART.

Several factors such as age, sex, and ethnicity, may impact Vδ2 cell profile distribution [18,36,37]. It has been previously reported that AA-uninfected individuals harbor a lower frequency of Vδ2 cells expressing CD56 and a higher frequency of cells primarily grouped within profile number 1 [17]. This profile is defined by a central memory phenotype associated with more proliferative and less cytotoxic responses [19]. Accordingly, we found that Vδ2 cells from AA PLWH grouped primarily in profile number 1. This observation may have critical therapeutic implications that warrant further investigation. An absence of information on the races of uninfected donors prevented extending our analysis to this cohort. Lastly, it has also been suggested that recovery of Vδ2 functionality is dependent upon the level of αβCD4 T cells [38]. We did not find a significant impact of CD4 counts on Vδ2 cell phenotypes, but we did detect a marginal effect associated with the frequency of total CD8 T cells.

In addition to depletion, HIV infection results in severe impairment of Vδ2 cell cytokine production and cytolytic capabilities [32,39]. This dysregulation is marked by a decrease in the secretion of proinflammatory cytokines IFN-γ and TNF-α as well as reduced killing capacity of Daudi cells. Previous studies on donors with variable viral suppression and time on ART have shown that prolonged ART may not completely restore these functions [40,41]. To our knowledge, our current study is the first to utilize a cohort of donors with complete viral suppression and a treatment duration of >1 year at the time of enrollment. We found that Vδ2 cells from PLWH are capable of cytokine production and display comparable killing capacity to uninfected donors. These results are in line with our previous work showing that PAM-expanded Vδ2 cells from ART-suppressed PLWH are capable of inhibiting viral replication and eliminating latently HIV-infected cells [22]. Although our measurement of cytokine production came from cultures with mixed PBMC populations [24], a positive correlation with CD56+ Vδ2 cells indicate they may represent the most abundant cytokine-producing population. Taken together, these data indicate that sustained ART is able to restore cytotoxic function of activated Vδ2 cell populations.

Although we have established that Vδ2 cell cytotoxic function is comparable between uninfected donors and PLWH who are fully suppressed for more than one year, there are some features that remain altered. In addition to low total circulating numbers, our results show lower TNF-α production along with higher IFN-γ and GrzB, most likely associated to HIV exposure. One possible explanation could be Vδ2 cells’ role as a reservoir for HIV [25]. While active HIV infection of γδ T cells does not appear to have a drastic impact on cytokine related function, it is unclear if integrated provirus alters their polyfunctionality [32]. Nevertheless, our observation of increased GrzB+ Vδ2 cells and higher IFN-γ upon ex vivo expansion fits with previous studies that suggest γδ T cells play a part in driving systemic inflammation that leads to HIV-associated comorbidities such as cardiovascular disease [42]. Alternatively, the combined increase in degranulation markers and IFN-γ may be indicative of some degree of HIV-specific Vδ2 cell responses that necessitate further investigation [22]. These findings add crucial information to the evaluation of ART and may guide the development of γδT cell immunotherapies aimed at curing persistent HIV infection

## Figures and Tables

**Figure 1 cells-09-02568-f001:**
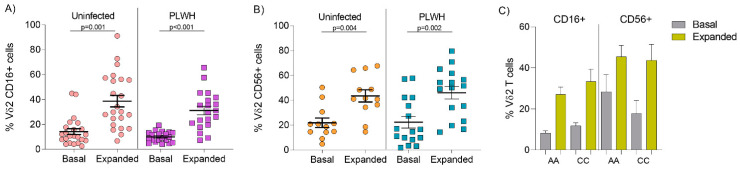
Expression of CD16 and CD56 on Vδ2 cells prior to and post PAM-expansion in uninfected donors and PLWH on ART. Comparable mean expression of basal (**A**) CD16 and (**B**) CD56 on Vδ2 cells between uninfected and PLWH and after seven days of exposure to PAM and IL-2, *p* = 0.42 and *p* = 0.43, respectively, Mann Whitney U-test. Both uninfected and PLWH increased CD16 and CD56 expression upon expansion (*p* < 0.001 and *p* < 0.001, respectively, Wilcoxon matched pairs signed rank test). (**C**) Similar distribution of basal and expanded mean CD16 expression on Vδ2 cells from nine African American (AA) and ten Caucasian (CC) PLWH donors (*p* = 0.08 and *p* = 0.60, respectively, Mann Whitney U-test). Comparable mean basal and expanded CD56 expression on Vδ2 cells in six AA and ten CC donors (*p* = 0.26 and *p* = 0.84, respectively, Mann Whitney U-test).

**Figure 2 cells-09-02568-f002:**
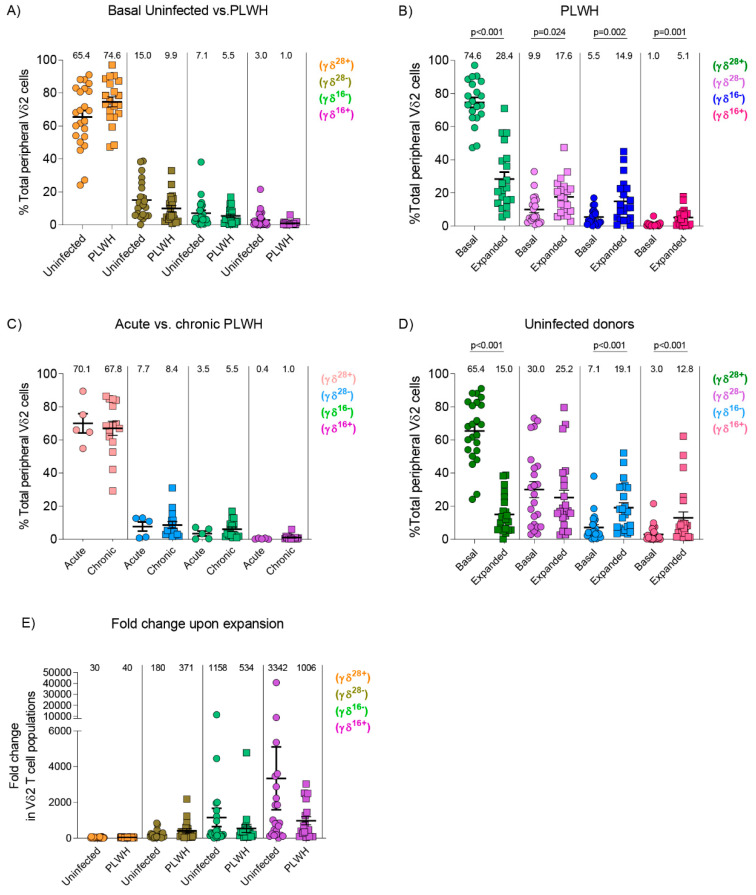
Vδ2 cell populations in uninfected and PLWH. γδ^(28+)^, γδ^(28−)^, γδ^(16+)^, and γδ^(16+)^ Vδ2 cell populations were defined as CD27+CD28+CD16- (γδ^(28+)^); CD27+CD28-CD16- (γδ^(28−)^); CD27-CD28-CD16- (γδ^(16+)^) and CD27-CD28-CD16+ (γδ^(16+)^), and analyzed in ex vivo (Basal) Vδ2 cells and after 7 days of expansion with PAM and IL-2 (Expanded) in 23 uninfected donors and 20 PLWH on ART. (**A**) Comparable ex vivo distribution of Vδ2 cell populations in uninfected donors and PLWH (*p* = 0.12, *p* = 0.07, *p* = 0.54, *p* = 0.40, respectively, Mann Whitney U test). (**B**) Distribution of basal and expanded Vδ2 cell subpopulations in PLWH. γδ^(28+)^ subpopulation decreased while γδ^(28−)^, γδ^(16−)^, and γδ^(16+)^ increased after expansion (Wilcoxon paired-signed rank test). (**C**) Comparable Vδ2 cell populations between PLWH treated in the acute or chronic phase of HIV infection (*p* = 0.63, *p* = 0.61, *p* = 0.67 and *p* = 0.77, respectively, Wilcoxon paired-signed rank test). (**D**) PAM expansion induced a decrease in the γδ^(28+)^ population, an increase in γδ^(16−)^ and γδ^(16+)^, while γδ^(28−)^ remained unchanged (*p* = 0.27) in uninfected donors (Wilcoxon matched-pairs signed rank test). (**E**) Fold-change expansion in uninfected donors and PLWH was comparable for all Vδ2 cell populations (*p* = 0.15, *p* = 0.07, *p* = 0.76, and *p* = 0.27, respectively, Mann Whitney U test). Numbers above the symbols in panels (**A**–**D**) represent the mean of each Vδ2 cell population, and in panel (**E**) represents the mean fold-change (expanded/basal) in Vδ2 cell populations.

**Figure 3 cells-09-02568-f003:**
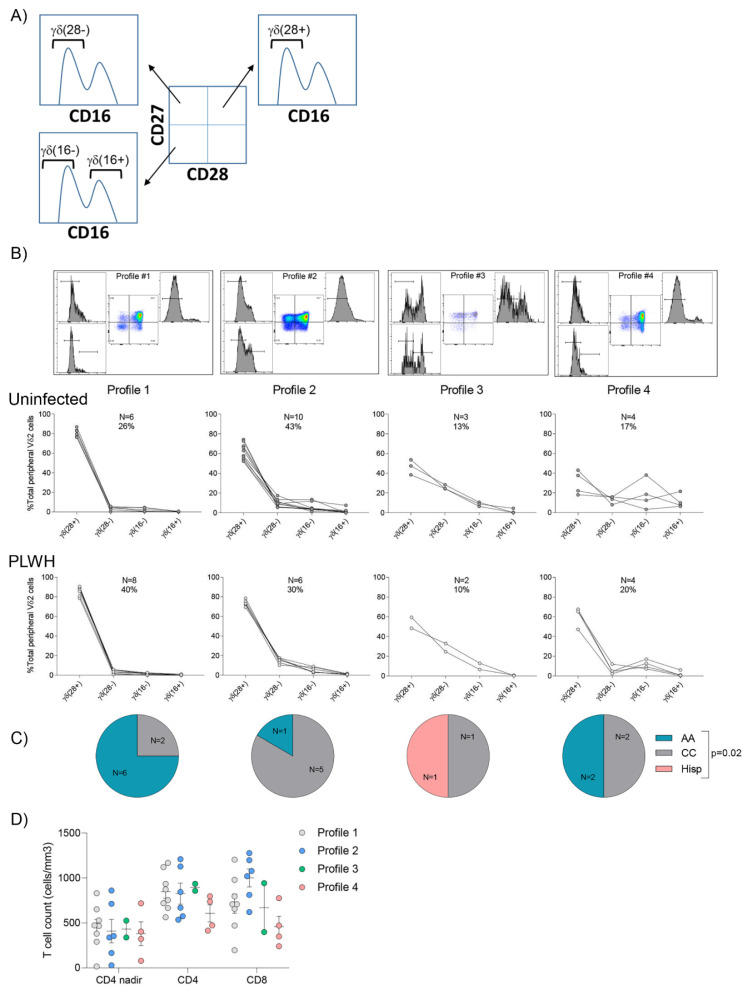
Classification of individuals according to the frequency of Vδ2 cell populations. Based on the classification of (**A**) four distinct Vδ2 populations (γδ^(28+)^, γδ^(28−)^, γδ^(16−)^, and γδ^(16+)^, individuals were grouped into four distinct profiles both in (**B**) Uninfected (N = 23, top pseudocolor plots showing representative plots of Vδ2 profile distribution, and top graphs) and PLWH (N = 20, bottom graphs). (**C**) Racial distribution within each profile amongst PLWH: African American (AA), Caucasian (CC), Hispanic (Hisp) (*p* = 0.02, Fisher exact test). (**D**) Classification of PLWH into the different profiles according to the frequency of nadir CD4 T cell count, CD4 T cell count and CD8 T cell count (cells/mm^3^) at the time of the visit (*p* = 0.95, *p* = 0.28 and *p* = 0.07, respectively, Kruskal-Wallis rank sum test).

**Figure 4 cells-09-02568-f004:**
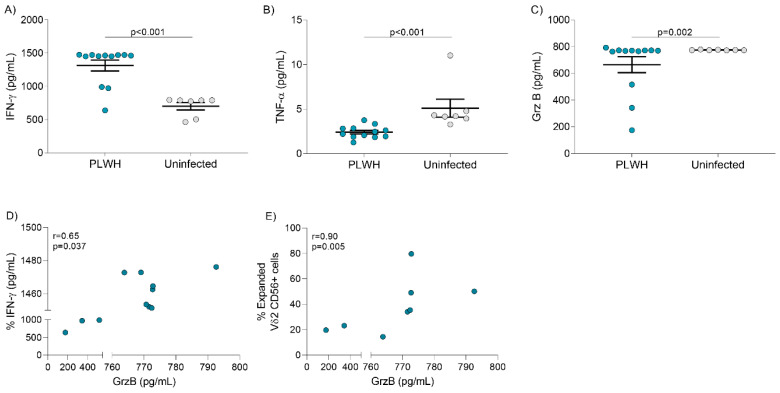
Cytokine release after seven days of expansion with PAM and IL-2. Quantification of cytokine levels was performed in supernatants from seven-day expansion cultures. Production of (**A**) IFN-γ (**B**) TNF-α and (**C**) GrzB in PLWH (N = 12) compared to uninfected donors (N = 7) (Mann-Whitney U-test). (**D**) Correlation between GrzB and IFN-γ released in the supernatant of expansion cultures (r = 0.65, *p* = 0.037, Spearman’s correlation test). (**E**) Correlation between GrzB released in the supernatant of expansion cultures and the frequency of Vδ2 CD56+ cells (r = 0.90, *p* = 0.005, Spearman’s correlation test).

**Figure 5 cells-09-02568-f005:**
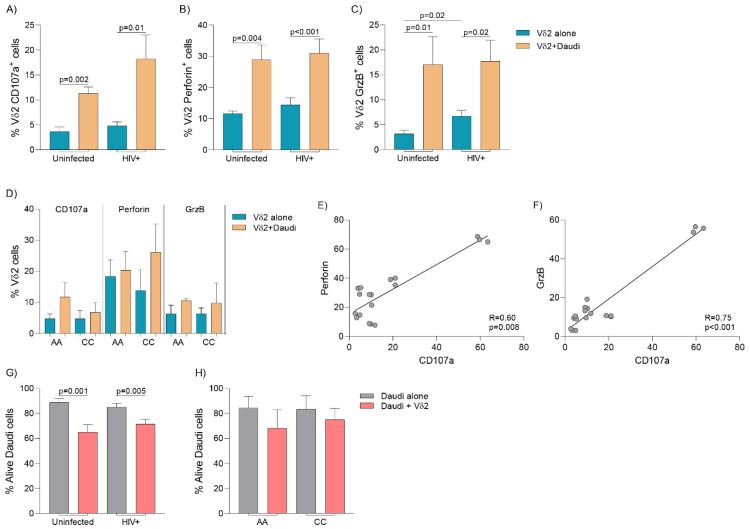
Cytolytic activity on expanded Vδ2 cells. Expanded Vδ2 cells were isolated and cocultured with Daudi cells (1:1 ratio) and degranulation and functional assays performed in uninfected donors (*n* = 4) and PLWH (*n* = 6). (**A**) Degranulation capacity was measured analyzing CD107a surface expression in isolated Vδ2 cells after expansion and upon coculture with Daudi cells. CD107a expression increased similarly in uninfected donors and PLWH (*p* = 0.81, Mann Whitney U-test). (**B**) Expanded Vδ2 cell intracellular production of perforin increased upon coculture with Daudi cells and was comparable between uninfected donors and PLWH (*p* = 0.97 and *p* = 0.60, respectively, Mann-Whitney U-test). (**C**) Expanded Vδ2 cells from PLWH produced more GrzB than uninfected donors (*p* = 0.02, Mann-Whitney U-test). Upon coculture with Daudi cells, similar GrzB production among uninfected donors and PLWH was measured (*p* = 0.34, Mann-Whitney U-test). (**D**) Degranulation capacity and production of perforin and GrzB was similar in African American (AA) and Caucasian (CC) PLWH (*p* > 0.05, Mann Whitney U-test). Positive correlation between (**E**) CD107a and perforin production, and (**F**) CD107a and GrzB (Spearman correlation test). (**G**) Vδ2 cell-mediated Daudi cell killing was comparable between uninfected donors and PLWH (*p* > 0.05, Mann Whitney U-test). (**H**) Comparable Vδ2-cell killing capacity between African American (AA) (*n* = 4) and Caucasian (CC) (*n* = 4) donors among PLWH (*p* > 0.05, Mann Whitney U-test).

**Figure 6 cells-09-02568-f006:**
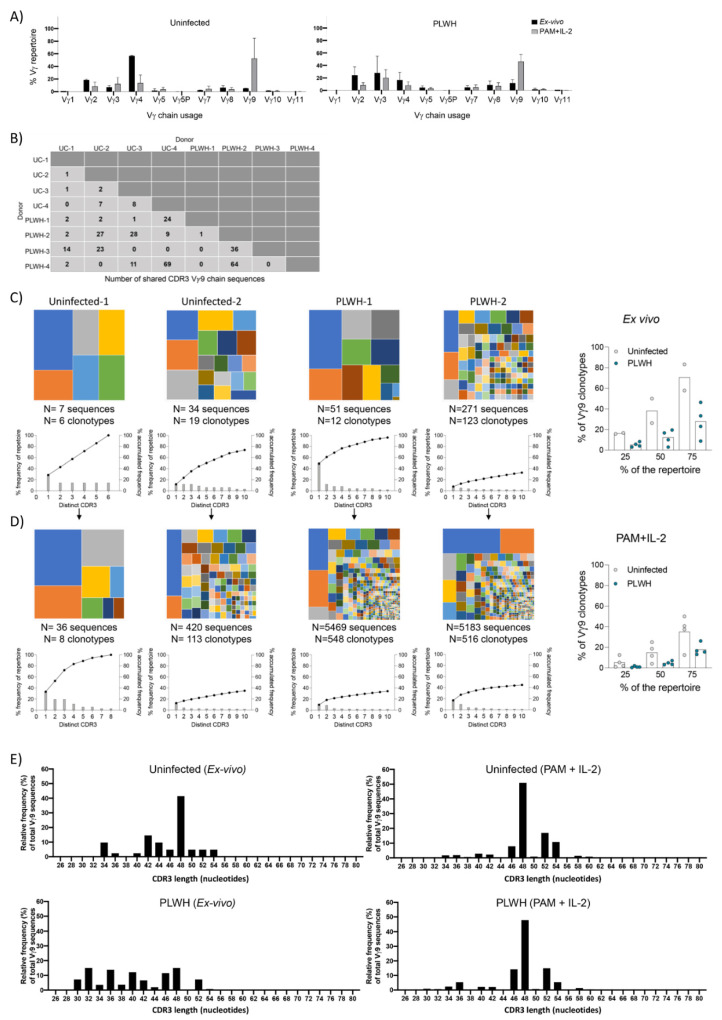
TCRγ chain analysis in uninfected donors compared to PLWH. (**A**) TCRγ chain usage exvivo (black) and after expansion (grey). (**B**) Publicity of Vγ9 sequences, showing the number of shared CDR3 Vγ9 sequences shared between the 8 individuals of the cohort. (**C**) (Left) Tree maps showing Vγ9 CDR3 clonotypes relative to repertoire size for uninfected and PLWH samples ex vivo with graphs below showing percentage clone abundance (grey bars) (left *y*-axis) for the top 10 most prevalent clonotypes. Accumulated frequency of the top 10 clonotypes is shown on the same graph (black line above bars). (Right) Graphs show the percentage of Vγ9 CDR3 clonotypes that make up various proportions of the total repertoire i.e., D25 (25%), D50 (50%), and D75 (75%). (**D**) Tree maps of uninfected and PLWH samples after 7-day treatment with PAM + IL-2. Representative PLWH samples are shown based on lowest and highest number of clonotypes. (**E**) Vγ9 CDR3 length from ex vivo and expanded samples from uninfected donors (UC) and PLWH. Ex vivo: UC (*n* = 2), PLWH (*n* = 4). Expanded: UC (*n* = 4), PLWH (*n* = 4).

**Table 1 cells-09-02568-t001:** Characteristics of PLWH on stable and suppressed ART.

N	20
Age (mean years)	42.3
Sex (% women)	5 (25%)
Race (%)	
African American	9 (45%)
Caucasian	10 (50%)
Hispanic	1 (2%)
Treated in the acute phase	5 (25%)
Median Time on ART (years)	6 (1.5–7.1)
Median CD4 nadir (cells/mm^3^)	392 (299–621)
Median CD4 count (cells/mm^3^)	776 (597–935)
Median CD8 count (cells/mm^3^)	742 (470–1010)

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
