# Peer review of "Comparable Vδ2 Cell Functional Characteristics in Virally Suppressed People Living with HIV and Uninfected Individuals"

_cells, 2020, doi:10.3390/cells9122568_

Round 1
Reviewer 1 Report
Dear Authors:
The manuscript entitled “Suppressive antiviral therapy rescues key Vd2 cell characteristics in people living with HIV” by M. L. Clohosey et al. presents a comparison of Vδ2 cell populations in the peripheral blood of uninfected donors and people living with HIV being treated with suppressive ART for > 1 year. The authors report that they found memory and effector phenotypes, post-expansion TCR repertoire diversity, and cytotoxic capability to be similar between the two groups in the study. Although there were many similarities in Vδ2 T-cell function between the two groups, (encouraging for ART therapy) there were differences in cytokine production and reduced proliferation of a cytotoxic effector population. The authors conclude that the information obtained in this study could be helpful in devising therapeutic strategies for HIV.
The great diversity of the human population is reflected in the data surrounding the isolated Vδ2 cells from the individuals included in the study. Because the study population is fairly low, 23 uninfected and 20 HIV infected, the standard deviations for the data sets are at times very high. This makes statistical analysis somewhat challenging. However, on the whole, the similarities between the Vδ2 cells from the uninfected controls and the HIV infected subjects are evident in the data presented for most of the experiments. For the most part the paper is well written, and the reported data make important contributions to the scientific knowledge regarding HIV and suppressive ART. I have several major comments and some minor points to make about the manuscript.
Major Points
- In line 113, it is mentioned that Ki-67 was among the markers used to characterize the Vδ2 T-cells. Staining with Ki-67 cannot be performed as indicated in this paragraph. The cells need to undergo permeabilization first before adding the Ki-67 stain, followed by several washes to remove unbound Ki-67. Then the cells are ready to be analyzed. This method paragraph needs to contain this information in a more explicit fashion.
- In line 133, it is mentioned that Daudi cells were chosen for the cytotoxicity experiments. Please give a short explanation for choosing this cell line. Also, it would be helpful to include in this manuscript representative examples of the flow cytometry results (dot plots and/or histograms) for the degranulation and cytotoxicity experiments. You could place these in the supplemental section for this manuscript.
- The paragraph (lines 251-266) discussing the results for the Vδ2 cell profiles needs more explanation about these profiles. How were they derived and how were they characterized? Also, more strongly state the importance of these profiles in the results section and the discussion section of the manuscript. As the paragraph is written now the reader needs to refer back to the original article about the profiles to understand how they were obtained and their importance.
- In addition, it would be helpful to include representative flow cytometry data for each of the Vδ2 profiles from PLWH subjects. These dot plots could be placed above the graphs for the profiles. In this way it will be more obvious to the reader how the numbers in the graphs were obtained.
- In figure 4 D and E it is difficult to see the high significance as indicated (r = .90 and p=0.005, etc.), since a regression line could not be inserted do to the break in the graph’s X and Y axis. Perhaps you can find a better way to present this data.
Minor Points
- In line 132 change “CD107a” to “CD107a(Lamp1)”
- In line 136, change “cell” to “cells”.
- In line 173, change “Figure S1B” to “Figure S1C”
- In line 174, change “Figure S1C” to “ Figure S1B”
- In lines 200 and 201 change the number 10 to the word ten in order to maintain consistency with the nine before AA and the six before AA.
- In lines 215 and 217 change the 10 to ten to maintain consistency with nine and six.
- In line 247, change “A-C” to “ A-D”
- In line 248, change “(D)” to “(E)”
- In line 265, delete the word “of”
Reviewer 2 Report
Based on their previous publications concerning subsets of Vg9d2 cells, this paper tested the hypothesis that despite continued depletion of Vδ2 cells during ART-treatment the phenotypic profiles, TCR repertoire, and cytotoxic capabilities of these subsets are “restored” to those observed in uninfected individuals. For this, comparisons of phenotype and function between uninfected individuals and people living with HIV (PLWH) on stable and suppressive ART for >1 year were performed, with outcomes supporting the hypothesis. In general, the paper is well written and technically sound.
Major critique:
1/ The claim that therapy restores the functional repertoire can only be made if a group of untreated HIV patients, or dynamic relation to onset of treatment is shown. For example, see ref 21 . Otherwise, the data shown, while valuable for understanding the Vg9Vd2 subset during ART are purely descriptive.
2/Section 3.5 . It would be relevant to show which cells in the cultures are producing the cytokines found in the supernatant.
3/ The authors stae “Although cell cultures contained 282 mixed PBMC populations, the expression of CD56 on expanded Vδ2 cells positively correlated with secreted GrzB levels (Figure 4E, p = 0.005) suggesting that the majority of this cytokine(?) was produced by Vδ2 cells. This may not be true as CD56 is expressed on NK cells which could likely produce Grzb. This issue should be addressed.
4/ In discussion they state Specifically, gd (16+) 432 cells in PLWH seemed to have an impaired capacity to expand compared to 433 uninfected donors. How is this supported by figure 2E ?
Round 2
Reviewer 2 Report
The authors have adequately addressed my concerns.